# The impact of removing financial incentives and/or audit and feedback on chlamydia testing in general practice: A cluster randomised controlled trial (ACCEPt-able)

Jane S. Hocking[1]*, Anna Wood[1,2], Meredith Temple-Smith[2], Sabine Braat[1], Matthew Law[3], Liliana Bulfone[4], Callum Jones[1], Mieke van Driel[5], Christopher K. Fairley[6,7], Basil Donovan[3], Rebecca Guy[3], Nicola Low[8], John Kaldor[3], Jane Gunn[9]

1 Melbourne School of Population and Global Health, University of Melbourne, Parkville, Victoria, Australia, 2 Department of General Practice, University of Melbourne, Parkville, Victoria, Australia, 3 Kirby Institute, University of New South Wales, Sydney, New South Wales, Australia, 4 College of Health and Medicine, Australian National University, Canberra, Australian Capital Territory, Australia, 5 Primary Care Clinical Unit, University of Queensland, Brisbane, Queensland, Australia, 6 Melbourne Sexual Health Centre, Melbourne, Victoria, Australia, 7 Alfred Hospital Clinical School, Monash University, Melbourne, Victoria, Australia, 8 Institute of Social and Preventive Medicine, University of Bern, Bern, Switzerland, 9 Faculty of Medicine, Dentistry and Health Sciences, University of Melbourne, Parkville, Victoria, Australia

* j.hocking@unimelb.edu.au

**Data Availability Statement:** Data cannot be shared publicly because ethics approval was not provided to make the data available. Ethics

## Abstract

### Background

Financial incentives and audit/feedback are widely used in primary care to influence clinician behaviour and increase quality of care. While observational data suggest a decline in quality when these interventions are stopped, their removal has not been evaluated in a randomised controlled trial (RCT), to our knowledge. This trial aimed to determine whether chlamydia testing in general practice is sustained when financial incentives and/or audit/feedback are removed.

### Methods and findings

We undertook a 2 × 2 factorial cluster RCT in 60 general practices in 4 Australian states targeting 49,525 patients aged 16–29 years for annual chlamydia testing. Clinics were recruited between July 2014 and September 2015 and were followed for up to 2 years or until 31 December 2016. Clinics were eligible if they were in the intervention group of a previous cluster RCT where general practitioners (GPs) received financial incentives (AU$5–AU$8) for each chlamydia test and quarterly audit/feedback reports of their chlamydia testing rates. Clinics were randomised into 1 of 4 groups: incentives removed but audit/feedback retained (group A), audit/feedback removed but incentives retained (group B), both removed (group C), or both retained (group D). The primary outcome was the annual chlamydia testing rate among 16- to 29-year-old patients, where the numerator was the number who had at least 1 chlamydia test within 12 months and the denominator was the number who had at

committee approval will be required for access to the original data from this trial. Inquiries about data access can be made to RACGP National Research and Evaluation Ethics Committee (ethics@racgp.org.au).

**Funding:** ACCEPt-able was funded by the National Health and Medical Research Council (NHMRC 1063597;named grant investigators included JSH, MTS, RG, JG, MvD, NL, LB; funder URL:https://www.nhmrc.gov.au/). JSH was supported by a NHMRC Senior Research Fellowship (1042907; https://www.nhmrc.gov.au/) and BD was supported by a NHMRC Practitioner Fellowship (1061035; https://www.nhmrc.gov.au/). The trial was investigator initiated. The funders had no role in study design, data collection and analysis, decision to publish, or preparation of the manuscript.

**Competing interests:** All authors have completed the ICMJE uniform disclosure form. I have read the journal's policy and the authors have the following competing interests: ML has received research grants to his institution from Gilead Sciences, Janssen-Cilag and ViiV Healthcare; NL is a member of the Editorial Board of PLOS Medicine.

**Abbreviations:** ACCEPt, Australian Chlamydia Control Effectiveness Pilot; GP, general practitioner; ICC, intra-cluster correlation coefficient; OR, odds ratio; RCT, randomised controlled trial.

least 1 consultation during the same 12 months. We undertook a factorial analysis in which we investigated the effects of removal versus retention of incentives (groups A + C versus groups B + D) and the effects of removal versus retention of audit/feedback (group B + C versus groups A + D) separately. Of 60 clinics, 59 were randomised and 55 (91.7%) provided data (group A: 15 clinics, 11,196 patients; group B: 14, 11,944; group C: 13, 11,566; group D: 13, 14,819). Annual testing decreased from 20.2% to 11.7% (difference −8.8%; 95% CI −10.5% to −7.0%) in clinics with incentives removed and decreased from 20.6% to 14.3% (difference −7.1%; 95% CI −9.6% to −4.7%) where incentives were retained. The adjusted absolute difference in treatment effect was −0.9% (95% CI −3.5% to 1.7%; $p = 0.2267$). Annual testing decreased from 21.0% to 11.6% (difference −9.5%; 95% CI −11.7% to −7.4%) in clinics where audit/feedback was removed and decreased from 19.9% to 14.5% (difference −6.4%; 95% CI −8.6% to −4.2%) where audit/feedback was retained. The adjusted absolute difference in treatment effect was −2.6% (95% CI −5.4% to −0.1%; $p = 0.0336$). Study limitations included an unexpected reduction in testing across all groups impacting statistical power, loss of 4 clinics after randomisation, and inclusion of rural clinics only.

## Conclusions

Audit/feedback is more effective than financial incentives of AU$5–AU$8 per chlamydia test at sustaining GP chlamydia testing practices over time in Australian general practice.

## Trial registration

Australian New Zealand Clinical Trials Registry ACTRN12614000595617

## Author summary

### Why was this study done?

- Financial incentives and audit/feedback are widely used in primary care to influence clinician behaviour and increase quality of care. As healthcare costs continue to increase, governments and funding agencies are reassessing funding models for primary care, with widespread cuts to financial incentives.

- While observational data suggest a decline in quality when these interventions are stopped, their removal has not been evaluated in a randomised controlled trial (RCT).

### What did the researchers do and find?

- We conducted a 2 × 2 factorial cluster RCT in Australian general practices that aimed to determine the impact on chlamydia testing in general practice when incentive payments per activity and/or audit/feedback on activity performance were removed.

- Clinics were randomised into 1 of 4 groups: incentives removed but audit/feedback retained, audit/feedback removed but incentives retained, both removed, and both retained.

- The primary outcome was the annual chlamydia testing rate among 16- to 29-year-old patients.

- We found that removal of incentive payments had little impact on general practice chlamydia testing, but the removal of audit and feedback reduced testing.

## What do these results mean?

- Our payments were consistent with other incentives general practitioners (GPs) received at the time, suggesting that in the Australian general practice setting, incentive payments of this amount do not have a substantial impact on influencing GP preventive healthcare activities such as chlamydia testing.

- The removal of quarterly audit and feedback for GPs had a greater impact on testing rates, reflecting the importance of this strategy in influencing GP preventive healthcare activities. The provision of audit and feedback was costlier than the provision of financial incentives. However, using online video conferencing and fully automating the audit and feedback reports would reduce costs.

- Our results suggest that, in Australia at least, audit and feedback is more effective than incentive payments of AU$5 to AU$8 per activity at influencing GP behaviour.

## Introduction

Primary care plays a fundamental role in preventive healthcare, and strategies to improve its quality include financial incentives and audit/feedback [1]. Financial incentives aimed at modifying provider behaviour to improve quality and/or increase efficiency in primary care [2] have been used by the Australian Government since 1998, when the Practice Incentives Program was introduced for activities such as diabetes care [3]. The program provides less than 10% of the funding for general practitioner (GPs) [4]. In the UK, the Quality and Outcomes Framework was introduced into the contract of GPs by the government in 2004, accounting for about 25% of primary care clinics' income [5]. Both schemes have been subject to debate about effectiveness [6–9] and have undergone modification, including withdrawal of some incentives and raising the payment threshold targets on others [5,10,11]. While some observational data suggest a decline in provider activities and quality of care when incentives are removed [5,12], other data have shown little impact [13,14]. There is little information about the effect of incentive removal on provider activities and quality of care in the Australian general practice setting. Further, the impact of the removal of incentives has not to our knowledge been assessed in a randomised controlled trial (RCT).

Audit/feedback is widely used in primary care [15–17]. In audit/feedback, GPs' professional practice is measured and compared with guidelines, targets, and/or peers, and results are fed back to the GPs. Ideally, this prompts them to modify their practice if the feedback finds this is

needed. While there is substantial RCT evidence that audit/feedback improves practice [18], observational data suggest that removing audit/feedback may reverse improvements. However, there is little evidence about the impact of removing audit/feedback on GP activities and quality of care in Australia, and to our knowledge no RCT evidence.

We had the unique opportunity to evaluate the impact of removing incentives and audit/feedback on the preventive activities of GPs in Australia by building on an existing trial—the Australian Chlamydia Control Effectiveness Pilot (ACCEPt) [19]. ACCEPt evaluated an intervention to increase chlamydia screening, a key preventive activity for young adults (<30 years) in Australian general practice [20]. The intervention included incentive payments for testing and audit/feedback on GPs' testing performance. At the end of ACCEPt, we re-randomised intervention clinics in a 2 × 2 factorial cluster RCT to determine whether preventive activities such as chlamydia testing in general practice are sustained when incentives and/or audit/feedback are removed. Given that the intention of financial incentives and/or audit/feedback is to modify provider behaviour in order to improve quality and/or increase efficiency, our hypothesis was that chlamydia testing would decrease if these strategies were removed. We present the results of this new trial, ACCEPt-able, here.

## Methods

ACCEPt-able was a 2 × 2 factorial cluster RCT and followed a published protocol [21]. We report the findings according to the CONSORT extension for cluster RCTs [22] (S1 CONSORT Checklist). There were no changes to trial recruitment, implementation, management, or follow-up methods, but in a change to the published protocol, we had to exclude clinics that were unable to provide outcome data at the end of the trial from the primary analysis (further detail provided below).

### Study design and participants

The parent trial, ACCEPt, was a cluster RCT that evaluated the effectiveness of a chlamydia screening intervention on chlamydia prevalence, finishing in December 2015. ACCEPt was conducted across 4 Australian states (New South Wales, Victoria, South Australia, and Queensland). Full details are published elsewhere [19,23]. At the time of ACCEPt, opportunistic chlamydia testing was recommended annually for sexually active 16- to 29-year-olds in general practice [20]. How chlamydia testing was conducted varied between clinics, with some clinics using GPs to initiate testing and others using practice nurses, and some using clinician-collected specimens for testing, others allowing patients to self-collect specimens (e.g., urine specimens or high vaginal swabs) and leave them at the clinic for testing, and others requiring the patient to attend an external pathology collection centre for testing. Intervention clinics received financial incentives to individual GPs for each chlamydia test: AU$5 per test for up to 20% of 16- to 29-year-olds tested each year to AU$8 per test for over 40% coverage. These payments were electronically transferred to the clinic each quarter. This amount was consistent with the payment of AU$6 GPs received at the time for completing immunisation schedules and corresponds to an annual payment of about AU$800 assuming an annual chlamydia testing rate of 20% and an average of 800 patients aged 16 to 29 years attending each clinic per year. This total amount, the payment frequency, and electronic transfer methods were consistent with those of other government-funded general-practice-based incentives at the time [4,24]. Intervention clinics also received audit/feedback, where individual GPs were provided with a 1-page report that summarised their chlamydia testing rates for the previous quarter, including the number of patients aged 16 to 29 years who had consulted them, the number they tested, and the number that tested positive. The report also included a statement of the

total amount of incentive payments they would receive for that quarter's testing. The report was given to individual GPs during a quarterly face-to-face visit with a research officer who explained the results and worked with the GP to identify strategies to help increase their testing rates. The intervention also included chlamydia education (hard-copy and online resources about chlamydia and its management that were given to all GPs and nurses in a face-to-face meeting with a research officer after randomisation) and computer alerts prompting testing. Not all clinics used the computer alerts. Guided by normalisation process theory, a member of the research team worked with each clinic to tailor the intervention to the resources of the clinic and to identify strategies to facilitate testing and embed it into routine practice [25]. Annual testing of 16- to 29-year-olds in intervention clinics increased from 8.2% to 20.1%, with a treatment effect odds ratio (OR) of 1.7 (95% CI 1.4 to 2.1) [19].

At the conclusion of ACCEPt, a research officer met with GPs in each intervention clinic, informed them about ACCEPt-able, invited them to participate, and obtained informed consent [21]. The intervention was allocated at the cluster level (clinic) because patients attending each clinic could consult with different GPs. Clinics were eligible if they were in the ACCEPt intervention arm. Patients aged 16–29 years were eligible for 1 chlamydia test per year unless they reported risk factors (e.g., new sex partner) or genital symptoms requiring further testing.

This trial was approved by the Royal Australian College of General Practitioners National Research and Evaluation Ethics Committee (NREEC 14–004; 16 May 2014), and written consent was obtained from all GPs. During ACCEPt-able, we recruited and consented new GPs, who were also provided with the chlamydia education package. Clinics were recruited into ACCEPt-able immediately after completing ACCEPt, between July 2014 and September 2015. Clinics were followed up for 2 years or until 31 December 2016, whichever came first.

### Randomisation and masking

Clinics were randomised using a computer-generated minimisation algorithm to maximise the balance across 2 variables—annual chlamydia testing rate among 16- to 29-year-olds in the clinic for 12 months prior to ACCEPt-able (<19% versus ≥19%, based on median testing rate) and number of 16- to 29-year-olds attending the clinic each year (<1,000 versus ≥1,000, based on the 67th percentile of the number of patients at each clinic, to ensure that groups were evenly distributed among relatively smaller and larger clinics because of the potential association of clinic size with patient quality of care [26]). The trial statistician was blinded to allocation. Blinding of clinics and GPs was not possible. Randomisation took place after clinics were recruited into ACCEPt-able and consented to participate. A research officer informed clinics and each GP of their allocation.

### Interventions

Clinics in ACCEPt-able were randomised into 1 of 4 arms: incentives removed but audit/feedback and visit retained (group A), audit/feedback and visit removed but incentives retained (group B), incentives and audit/feedback and visit removed (group C), or incentives and audit/feedback and visit retained (group D). All GPs within each clinic received the same intervention. The groups receiving audit/feedback received the same quarterly 1-page report as for the ACCEPt trial that summarised GPs' chlamydia testing rate for the previous quarter and included a statement of the total amount of incentive payments they would receive for that quarter's testing. The report was given during a quarterly face-to-face visit with a research officer who explained the results and worked with GPs to identify strategies to help increase their testing rates.

## Outcomes

The primary outcome was annual chlamydia testing rate among 16- to 29-year-olds attending the clinic. The numerator was the number of patients aged 16–29 years who had at least 1 chlamydia test within 12 months; the denominator was the number of patients aged 16–29 years who had at least 1 consultation during the same 12 months.

Testing data were extracted from each clinic's electronic medical records using GRHANITE [27,28], a data extraction tool. The tool extracts consultation data including a unique non-identifying patient code, the age and sex of the patient, and chlamydia test results. Data were extracted for the 12 months prior to commencement in ACCEPt-able and during the intervention period.

## Sample size

The sample size was determined by ACCEPt, which included 60 intervention clinics. We had 94% power to detect a 5% absolute decrease in annual chlamydia testing from 20% to 15% between any 2 groups. A 5% reduction represents a clinically relevant result—about 200,000 fewer 16- to 29-year-olds screened each year in Australia. Our calculations assumed an intra-cluster correlation coefficient (ICC) of 0.02 for testing rate [19], an average cluster size of 700 patients aged 16–29 years per clinic per year, and an alpha of 0.05.

## Statistical analysis

We conducted a factorial analysis as our primary analysis. This investigated the effects of removal versus retention of incentives (groups A + C versus groups B + D) and audit/feedback (groups B + C versus groups A + D) separately on annual chlamydia testing over 2 years. We aimed to compare the groups according to intention-to-treat, but in a change to the published protocol [21], we had to exclude clinics that were unable to provide outcome data at the end of the trial from the primary analysis. For each intervention, we fitted generalised linear models, using generalised estimating equations to account for clustering at the clinic level, and assessed the impact of the intervention on chlamydia testing in year 2 compared with baseline. A logistic model generated ORs, and absolute differences were obtained from a model with an identity link function with binomial error distribution. These models also provided 95% confidence intervals and p-values and adjusted for minimisation variables only (annual chlamydia testing rate among 16- to 29-year-olds in the clinic and number of 16- to 29-year-olds attending the clinic each year), as is recommended [29]. We also obtained the results of an adjusted model post hoc that, in addition to the minimisation factors, also included the variables that were adjusted for in the ACCEPt trial (patient sex and age group and socio-economic status quintile of the clinic—'fully adjusted model') [19,30].

We undertook several post-hoc analyses: (i) we calculated absolute differences in addition to the planned ORs; (ii) we tested the assumption that there was no interaction effect between the 2 interventions and conducted an analysis by randomised group whereby the group that retained audit/feedback and incentives was the control ('intervention group analysis'), as is recommended for reporting factorial trials [31]; (iii) we calculated the ICC for chlamydia testing using the primary analysis model with trial arm in the model; and (iv) we conducted factorial subgroup analyses by sex and age group (16–19, 20–24, and 25–29 years). The output was generated using SAS software, version 9.4, for Windows.

## Cost–consequence analysis

A cost–consequence analysis comparing costs and consequences for each combination of removing/retaining incentives and audit/feedback activities was conducted [32]. Costs

(incentives, travel, staff time, and data extraction) and consequences (proportion of the target population tested) for the scenarios of removing versus retaining each intervention were obtained from trial data. The average saving per patient aged 16–29 years was calculated for removal of each intervention. The incremental cost of retaining each intervention per additional patient in the target population tested was calculated. As the trial was based in rural clinics, we conducted a sensitivity analysis to examine the potential costs and consequences for removing or retaining the interventions in metropolitan clinics, where travel costs and staff time for travel are likely to be reduced considerably.

## Results

Of 60 clinics, 59 agreed to participate in ACCEPt-able. No clinics withdrew, but 4 clinics had technical problems with data extraction and their data were unavailable, leaving 55 (91.7%) clinics in the analysis (Fig 1). The intervention period ranged from 0.2 years to 2 years, with a mean duration of 1.5 years (SD 0.4). Three clinics participated for less than 1 year (2 clinics closed and 1 clinic was a solo GP who became unwell and ceased seeing patients), 23 clinics between 1 and 1.5 years, and 29 clinics between 1.5 and 2 years. The average duration of the intervention period was similar between groups (1.5 years for groups A and C; 1.6 years for groups B and D).

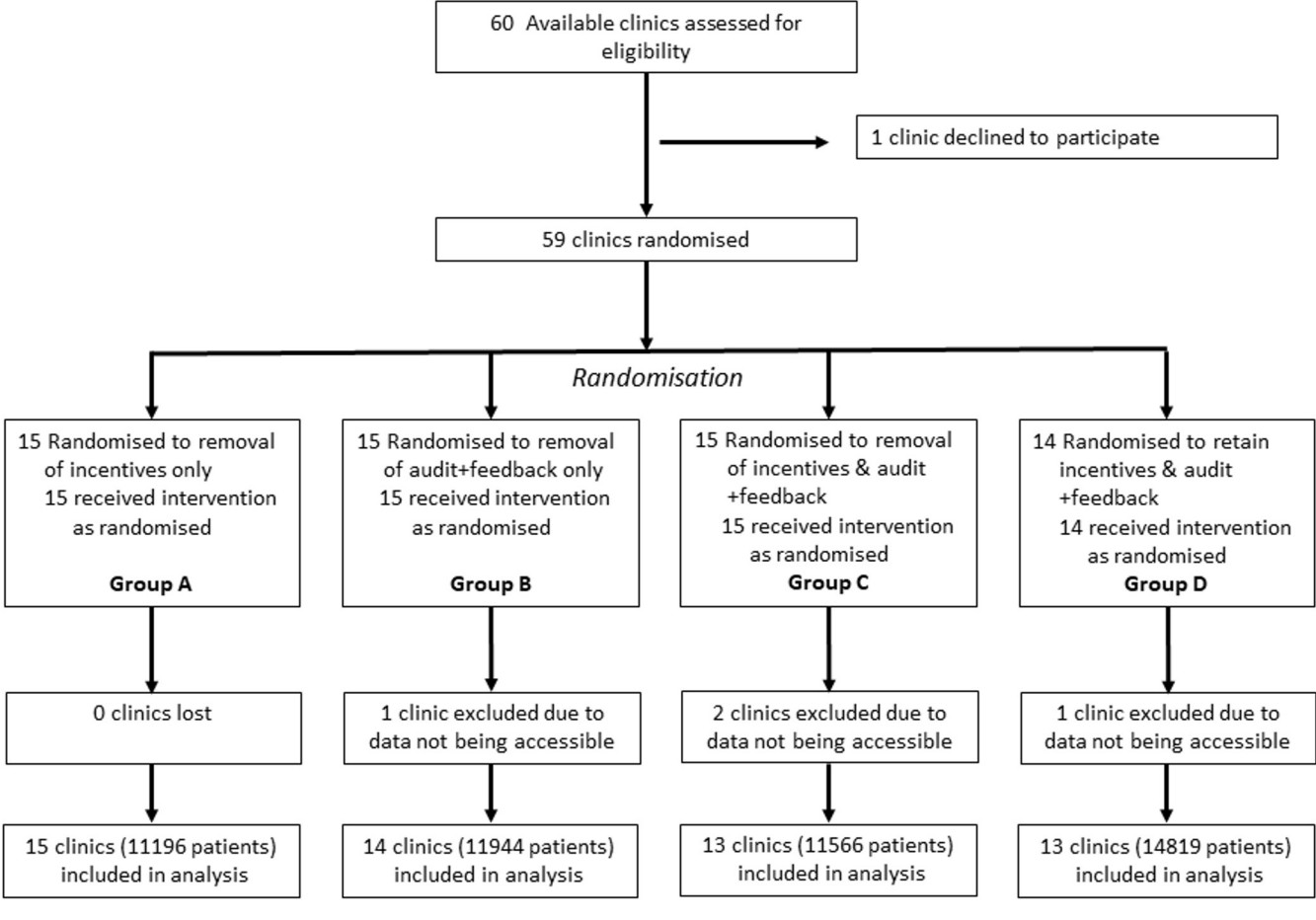

**Fig 1. Flow chart.**

Baseline characteristics at the patient and cluster level were similar between pairs of intervention groups (for factorial analysis) (Table 1), but given the loss of 4 clinics, we report only the results from the fully adjusted models in the text. The results from the model adjusted for minimisation variables only and the results from the fully adjusted model (adjusted for minimisation variables and patient age and sex and socio-economic status of the clinic) were similar (Table 2). There were some minor differences between the 4 trial groups, with clinics in group C (incentives and audit/feedback removed) and group D (incentives and audit/feedback retained) being more likely to be in disadvantaged areas. For analyses reporting on each intervention group ('intervention group analysis'), we report the fully adjusted analyses.

Chlamydia testing rates decreased from baseline in all groups (Figs 2–4), and for groups A, B, and C, testing rates reduced to levels like those observed in the first 12 months of ACCEPt, the parent trial (S1 Fig).

There was no statistical evidence of an interaction for treatment effect between removal of incentives and removal of audit/feedback on our primary outcome of chlamydia testing

**Table 1. Baseline characteristics of clinics and patients.**

| Characteristic | Total sample | Pairs of intervention groups[a] | | | | Intervention groups[b] | | | |
|---|---|---|---|---|---|---|---|---|---|
| | | Incentives removed (A + C) | Incentives retained (B + D) | Audit/feedback removed (B + C) | Audit/feedback retained (A + D) | Removal of incentives only (A) | Removal of audit/feedback only (B) | Removal of both incentives and audit/feedback (C) | Control—incentives and audit/feedback retained (D) |
| **Clinic-level characteristics** | | | | | | | | | |
| Number of clinics | 55 | 28 | 27 | 27 | 28 | 15 | 14 | 13 | 13 |
| Socio-economic status of the clinic location, n (%)[c] | | | | | | | | | |
| Q1 (most disadvantaged) | 12 (21.8) | 5 (17.9) | 7 (25.9) | 7 (25.9) | 5 (17.9) | 0 (0.0) | 2 (14.3) | 5 (38.5) | 5 (38.5) |
| Q2 | 35 (63.6) | 19 (67.9) | 16 (59.3) | 17 (63.0) | 18 (64.3) | 12 (80.0) | 10 (71.4) | 7 (53.8) | 6 (46.1) |
| Q3 | 4 (7.3) | 2 (7.1) | 2 (7.4) | 1 (3.7) | 3 (10.7) | 1 (6.7) | 0 (0.0) | 1 (7.7) | 2 (15.4) |
| Q4 | 3 (5.5) | 2 (7.1) | 1 (3.7) | 1 (3.7) | 2 (7.1) | 2 (13.3) | 1 (7.1) | 0 (0.0) | 0 (0.0) |
| Q5 (least disadvantaged) | 1 (1.8) | 0 (0.0) | 1 (3.7) | 1 (3.7) | 0 (0.0) | 0 (0.0) | 1 (7.1) | 0 (0.0) | 0 (0.0) |
| Number of GPs, n (IQR of number of GPs per clinic) | 383 (6[3–9]) | 195 (6[2–8]) | 188 (6[3–9]) | 185 (6[2–10]) | 198 (6[3–9]) | 103 (6[4–7]) | 93 (5[3–9]) | 92 (6[2–10]) | 95 (6[3–7]) |
| **Patient-level characteristics** | | | | | | | | | |
| Number of patients in the 12 months prior to randomisation[d] | 49,525 | 22,762 | 26,763 | 23,510 | 26,015 | 11,196 | 11,944 | 11,566 | 14,819 |
| Patient age group, n (%) | | | | | | | | | |
| 16–20 years | 15,205 (30.7) | 6,985 (30.7) | 8,220 (30.7) | 7,202 (30.6) | 8,003 (30.8) | 3,525 (31.5) | 3,742 (31.3) | 3,460 (29.9) | 4,478 (30.2) |
| 20–24 years | 17,564 (35.5) | 8,047 (35.3) | 9,517 (35.6) | 8,285 (35.2) | 9,279 (35.7) | 3,906 (34.9) | 4,144 (34.7) | 4,141 (35.8) | 5,373 (36.3) |
| 25–29 years | 16,756 (33.8) | 7,730 (34.0) | 9,026 (33.7) | 8,023 (34.1) | 8,733 (33.6) | 3,765 (33.6) | 4,058 (34.0) | 3,965 (34.3) | 4,968 (33.5) |
| Patient sex, n (%)[d] | | | | | | | | | |
| Male | 20,589 (41.6) | 9,623 (42.3) | 10,966 (41.0) | 10,093 (42.9) | 10,496 (40.3) | 4,726 (42.2) | 5,196 (43.5) | 4,897 (42.3) | 5,770 (38.9) |
| Female | 28,936 (58.4) | 13,139 (57.7) | 15,797 (59.0) | 13,417 (57.1) | 15,519 (59.7) | 6,470 (57.8) | 6,748 (56.5) | 6,669 (57.7) | 9,049 (61.1) |
| Chlamydia testing rate in the 12 months prior to randomisation, n/N (%) | 10,109/49,525 (20.4) | 4,592/22,762 (20.2) | 5,517/26,763 (20.6) | 4,935/23,510 (21.0) | 5,147/26,015 (19.9) | 2,124/11,196 (19.0) | 2,467/11,944 (20.6) | 2,468/11,566 (21.3) | 3,050/14,819 (20.6) |

n = number tested aged 16 to 29 years; N = number of individuals aged 16 to 29 years attending the clinic.

[a]For factorial analysis.

[b]For intervention group analysis.

[c]Socio-economic status is based on quintiles (Q) of the Socio-Economic Indexes for Areas (SEIFA) Index of Relative Socio-economic Disadvantage (IRSD) of the Australian Bureau of Statistics 2011 census for the postcodes of each clinic location.

[d]Number of patients aged 16 to 29 years attending participating clinics in the 12-month period prior to randomisation.

GP, general practitioner; IQR, interquartile range.

**Table 2. Primary outcome chlamydia testing—factorial analysis.**

**Impact of removal of incentive payments**

| Time point or outcome | Incentive payments removed (groups A + C) (intervention) | | Incentive payments retained (groups B + D) (control) | | Treatment effect[a] | | Adjusted treatment effect[b] | |
|---|---|---|---|---|---|---|---|---|
| | n/N | Testing rate, percent (95% CI) | n/N | Testing rate, percent (95% CI) | OR (95% CI) | p-Value | OR (95% CI) | p-Value |
| Baseline[c] | 4,592/ 22,762 | 20.2 (18.2 to 22.1) | 5,517/ 26,763 | 20.6 (18.2 to 23.0) | 0.9 (0.8 to 1.1) | 0.4567 | 1.0 (0.9 to 1.1) | 0.4729 |
| Year 1[c] | 3,032/ 21,284 | 14.2 (12.6 to 15.9) | 4,292/ 26,752 | 16.0 (12.9 to 19.2) | 0.8 (0.7 to 1.0) | 0.0755 | 0.9 (0.7 to 1.0) | 0.1017 |
| Year 2[c] | 1,720/ 14,651[d] | 11.7 (9.9 to 13.6) | 3,009/ 21,076[b] | 14.3 (10.3 to 18.2) | 0.8 (0.6 to 1.1) | 0.1039 | 0.8 (0.6 to 1.1) | 0.1774 |
| Year 2 versus baseline (95% CI)[a] | — | Diff: −8.8 (−10.5 to −7.0) OR: 0.5 (0.4 to 0.6) | — | Diff: −7.1 (−9.6 to −4.7) OR: 0.6 (0.5 to 0.8) | — | — | — | — |
| Treatment effect (removal-retain) (95% CI) | — | — | — | — | Diff: −1.6 (−4.6 to 1.3) OR: 0.8 (0.6 to 1.1) | 0.1964 | Diff: −0.9 (−3.5 to 1.7) OR: 0.8 (0.6 to 1.1) | 0.2267 |

**Impact of removal of audit/feedback**

| Time point or outcome | Audit/feedback removed (groups B + C) (intervention) | | Audit/feedback retained (groups A + D) (control) | | Treatment effect[a] | | Adjusted treatment effect[b] | |
|---|---|---|---|---|---|---|---|---|
| | n/N | Testing rate% (95% CI) | n/N | Testing rate% (95% CI) | OR (95% CI) | p-Value | OR (95% CI) | p-Value |
| Baseline[c] | 4,935/ 23,510 | 21.0 (18.8 to 23.2) | 5,147/ 26,015 | 19.9 (17.6 to 22.1) | 1.0 (0.9 to 1.2) | 0.6674 | 1.0 (0.9 to 1.2) | 0.7514 |
| Year 1[c] | 3,329/ 22,738 | 14.6 (12.7 to 16.5) | 3,995/ 25,298 | 15.8 (12.6 to 19.0) | 0.9 (0.7 to 1.1) | 0.2010 | 0.9 (0.7 to 1.0) | 0.1293 |
| Year 2[c] | 1,809/ 15,643[d] | 11.6 (9.4 to 13.8) | 2,920/ 20,084[b] | 14.5 (10.6 to 18.5) | 0.7 (0.5 to 1.0) | 0.0882 | 0.7 (0.5 to 0.9) | 0.0191 |
| Year 2 versus baseline (95% CI)[a] | — | Diff: −9.5 (−11.7 to −7.4) OR: 0.5 (0.4 to 0.6) | — | Diff: −6.4 (−8.6 to −4.2) OR: 0.6 (0.5 to 0.8) | — | — | — | — |
| Treatment effect: (removal-retain) (95% CI) | — | — | — | — | Diff: −3.1 (−6.2 to −0.1) OR: 0.7 (0.5 to 1.0) | 0.0374 | Diff: −2.6 (−5.4 to −0.2) OR: 0.7 (0.5 to 1.0) | 0.0336 |

n = number tested aged 16 to 29 years; N = number of individuals aged 16 to 29 years attending the clinic.

[a]Models account for minimisation variables including annual chlamydia testing rate among 16- to 29-year-olds and number of 16- to 29-year-olds attending the clinic each year.

[b]The fully adjusted model contains patient sex, patient age group, and socio-economic status of the clinic (continuous) in addition to the minimisation variables.

[c]Baseline is the 12-month period prior to randomisation. Year 1 is 1–12 months after randomisation. Year 2 is 13–24 months after randomisation.

[d]Numerator and denominator less than for baseline and year 1 because not all clinics contributed 12 months of data to year 2.

Diff, absolute difference; OR, odds ratio.

(interaction effect = 3.2%; 95% CI −2.4% to 8.8%; p = 0.2642). The ICC for testing was 0.015. In our factorial analysis, the annual chlamydia testing rate decreased from 20.2% to 11.7% over the 2 years (difference −8.8%; 95% CI −10.5% to −7.0%) where incentives were removed and decreased from 20.6% to 14.3% (difference −7.1%; 95% CI −9.6% to −4.7%) where incentives were retained. The adjusted absolute difference in treatment effect between groups was −0.9% (95% CI −3.5% to 1.7%; p = 0.2267), and the adjusted OR was 0.8 (95% CI 0.6 to 1.1; p = 0.2267) (Table 2). In subgroup analyses, the differences in treatment effect between clinics where incentives were removed and clinics where incentives were retained when stratified by sex or age of patient were not statistically significant (S1 Table). Annual testing decreased from

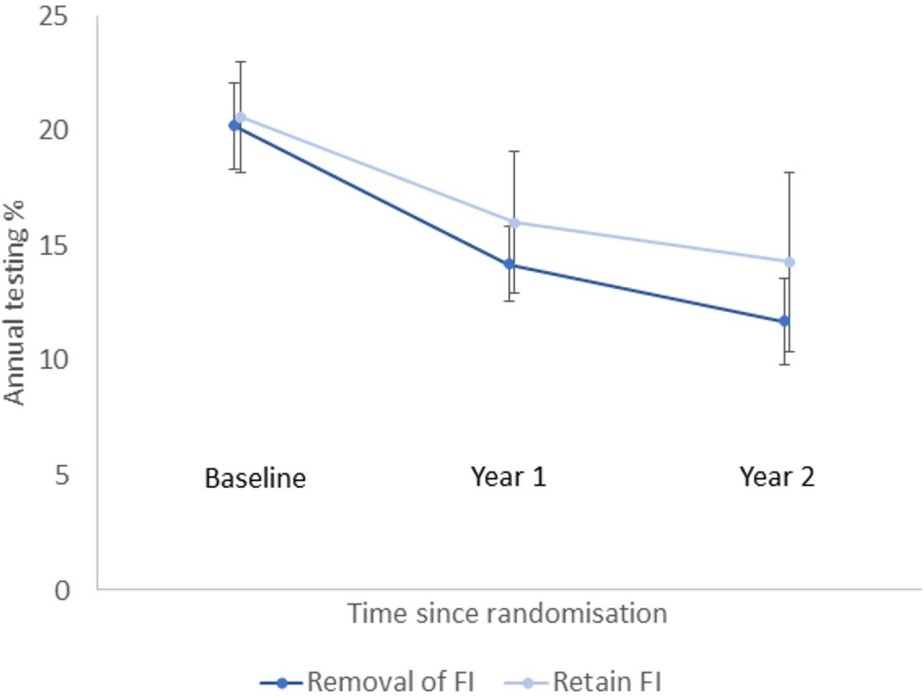

**Fig 2. Proportion of patients tested for chlamydia per year by time since randomisation: Factorial analysis—removal of financial incentives versus retention of financial incentives.** Error bars correspond to 95% confidence intervals. FI, financial incentives.

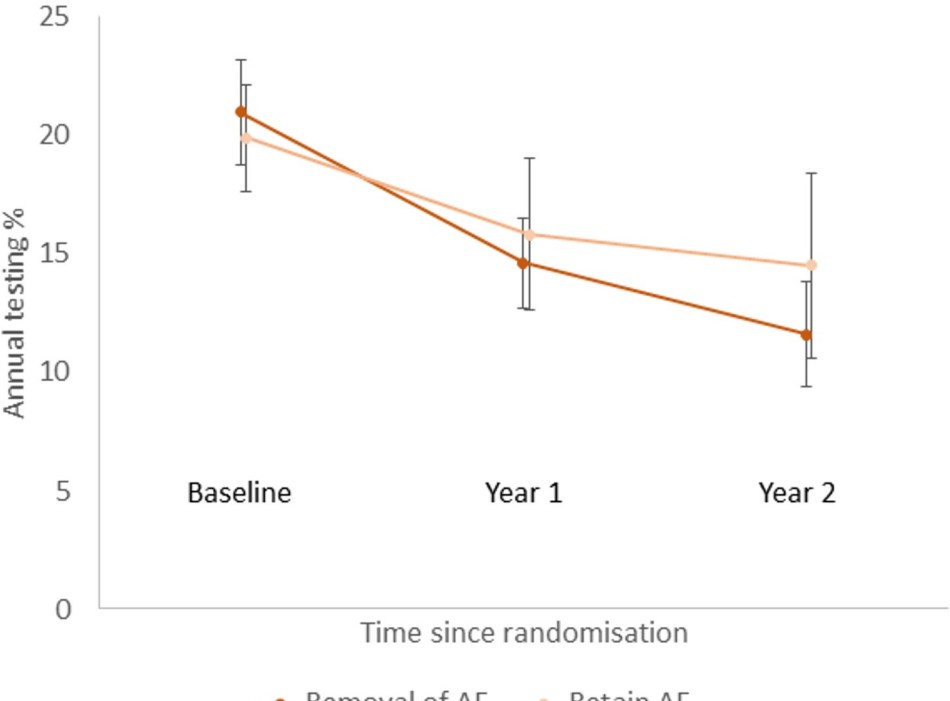

**Fig 3. Proportion of patients tested for chlamydia per year by time since randomisation: Factorial analysis—removal of audit/feedback versus retention of audit/feedback.** Error bars correspond to 95% confidence intervals. AF, audit/feedback.

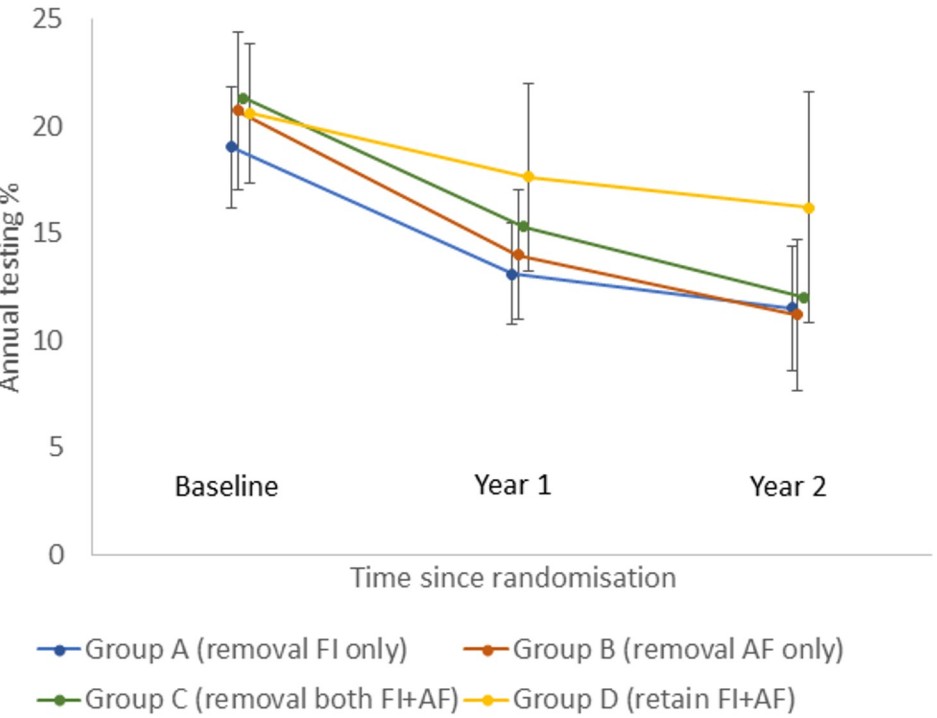

**Fig 4. Proportion of patients tested for chlamydia per year by time since randomisation: Intervention group analysis.** Error bars correspond to 95% confidence intervals. AF, audit/feedback; FI, financial incentives.

21.0% to 11.6% over the 2 years (difference −9.5%; 95% CI −11.7% to −7.4%) where audit/feedback was removed and decreased from 19.9% to 14.5% (difference −6.4%; 95% CI −8.6% to −4.2%) where audit/feedback was retained. The adjusted absolute difference in treatment effect was greater for removal than retention of audit/feedback (difference −2.6%; 95% CI −5.4% to −0.2%; $p = 0.0336$), and the adjusted OR was 0.7 (95% CI 0.5 to 1.0; $p = 0.0336$) (Table 2). In subgroup analyses, evidence of a difference was observed when stratified by sex and age group of patients except for those aged 25 to 29 years (S1 Table). The absolute difference in treatment effect did not vary between age groups.

Our intervention group analysis showed that testing decreased in all 4 groups, but the decrease was substantially lower in the group that retained incentives and audit/feedback. The adjusted absolute treatment effects were −1.8% (95% CI −4.9% to 1.3%; $p = 0.0660$) for removal of incentives only, −3.4% (95% CI −7.8% to 1.0%; $p = 0.0247$) for removal of audit/feedback only, and −3.4 (95% CI −6.5% to −0.2%; $p = 0.0356$) for removal of incentives and audit/feedback (S2 Table).

## Cost and consequences

There was an estimated cost saving of AU\$2.31 per 16- to 29-year-old patient per year associated with removing incentives. As removal of incentives had no significant impact on testing, discontinuing incentives dominates over a strategy of their retention (Table 3). There was an estimated cost-saving of AU\$5.88 per 16- to 29-year-old patient per year associated with removing audit/feedback. The incremental cost of continuing audit/feedback activities was an estimated AU\$189.64 (range: AU\$94.82 to AU\$5,117.49) per additional patient in the target population tested (Table 3). Most costs for audit/feedback were travel-related (79%). Sensitivity analysis showed that if travel costs were reduced to reflect the costs for research officers to

visit metropolitan clinics, the costs of audit/feedback would decrease to an average of AU$3.02 per patient (Table 3).

## Discussion

In a 2 × 2 factorial cluster RCT set in Australian general practice, the removal of financial incentives of AU$5 to AU$8 paid to GPs for each chlamydia test conducted had little additional impact on reducing testing rates among 16- to 29-year-olds attending the clinic. Our payments were consistent with other incentives at the time [24], suggesting that in the Australian general practice setting, incentives at this level do not have an important impact on preventive activities like chlamydia testing. We found that the removal of audit/feedback reduced testing, with a relative reduction of 30% (absolute difference = −2.6%) that could translate to about 160,000 fewer 16- to 29-year-olds tested each year in Australia [33]. The provision of audit/feedback was costlier, but most costs were for the visit, which could be substantially reduced with online conferencing for example. Fully automating the audit and feedback reports using digital platforms would also further reduce costs. We also found that chlamydia testing rates declined in all groups, regardless of whether incentives and/or audit and feedback were removed, emphasising the challenges in sustaining preventive healthcare activities in general practice over time.

There are several explanations for why we did not see an impact of removal of incentives. Incentives may not have been critical in driving test uptake in ACCEPt, such their removal in ACCEPt-able did not substantially impact testing. At the beginning of ACCEPt-able, clinics received an average total payment of AU$822 per year for chlamydia testing, which, at the time, was consistent with the total amount of approximately AU$2,400 that clinics received across 3 activities (asthma and diabetes cycles of care and cervical screening in under-screened women) as part of the Practice Incentives Program [34]. The introduction of these incentives in 2001 did not significantly increase uptake of these activities, suggesting incentivisation like this is unlikely to translate into substantial changes in Australian general practice [4]. This is supported by qualitative research, where Australian GPs report that incentives do not fundamentally influence patient management [4,35]. This may be because Australian general practices are largely funded by a fee-for-service reimbursement model; the few incentives available represent less than 10% of their funding [4]. Chance cannot be excluded because we did not expect a reduction in testing in clinics that retained incentives, which reduced our effective sample size, and our observed treatment effect of 0.9% was considerably smaller than our hypothesized 5%.

Our audit/feedback intervention included a written report and visit by a research officer. Unfortunately, we could not determine whether removing the report or the visit alone would have had the same effect. However, a previous systematic review compared an educational visit plus audit/feedback with audit/feedback alone, finding that the 2-pronged approach was more effective than audit/feedback only [36].

Unexpectedly, we observed that testing also decreased in the group that retained incentives and audit/feedback. This suggests that chlamydia testing had not become normalised in work practices, with clinics returning to their pre-intervention ways of working despite the intervention's remaining in place [25]. Alternatively, it is possible that staff turnover led to loss of 'corporate memory' [37] about chlamydia, contributing to reduced testing. We provided clinics with the same level of support during ACCEPt-able as during ACCEPt, but we did not monitor whether there were changes in the clinics' use of other strategies to facilitate testing such as using computer alerts, and while new GPs received our chlamydia educational package, we did not provide any further educational support to already-participating GPs. The lack of ongoing

**Table 3. Cost and consequences evaluation.**

| Variable | ACCEPt-able costs (rural clinics) | | | Estimated costs for metro clinics (sensitivity analysis) | | |
|---|---|---|---|---|---|---|
| | Average number of hours used per clinic per quarter (range) | Hourly cost (in AUD) | Average total cost (AUD) per clinic per quarter (range) (unless otherwise indicated) | Average number of hours used per clinic per quarter (range) | Hourly cost (in AUD) | Average total cost (AUD) per clinic per quarter (range) (unless otherwise indicated) |
| **Incentive payments** | | | | | | |
| **Program-related activities and resources: Labour and technical support to collate data for dispensing incentive payments to clinics** | | | | | | |
| Analysis of medical record data (includes extraction, parsing, and cleaning of data to generate report of incentive payments due) | | | $250.00 ($225.00, $312.50) | | | $250.00 ($225.00, $312.50) |
| Staff administration of incentive payments | 0.25 (0.13, 0.5) | $57.50[a] | $14.38 ($7.48, $57.50) | 0.25 (0.13, 0.5) | $57.50[a] | $14.38 ($7.48, $57.50) |
| **Total and incremental costs** | | | | | | |
| Total costs per clinic per quarter to authorise payments | | | $264.38 ($232.48, $370.00) | | | $264.38 ($232.48, $370.00) |
| Total costs for 28 clinics per year to authorise payments | | | $29,610 | | | $29,610 |
| Total incentive payments for 28 clinics at 20.2% testing rate per year[b] | | | $23,006 | | | $23,006 |
| Average incentive payments per clinic per year | | | $822 | | | $822 |
| Total costs for 28 clinics per year to provide incentives | | | $52,616 | | | $52,616 |
| Total reduction in costs for removal of incentives | | | −$52,616 | | | −$52,616 |
| Total reduction in costs per clinic per year for removal incentives | | | −$1,879 | | | −$1,879 |
| Number of people in the target population in the 28 clinics where incentives were removed[b] | | | 22,762 | | | 22,762 |
| Average saving per patient per year in the target population for removal of incentives | | | −$2.31 | | | −$2.31 |
| Incremental change in proportion of target patients tested through removal of incentives (95% CI)[c] | | | −1.6% (−4.6%, 1.3%) | | | −2.1% (−5.6%, 1.4%) |
| Incremental cost of incentive payments per additional patient per year in the target population tested (range) | | | Dominant | | | Dominant |
| **Audit and feedback** | | | | | | |
| **Program-related activities and resources: Labour and technical support to collate data and generate audit report** | | | | | | |
| Preparation of medical record reports (includes extraction, parsing, and cleaning of data and generating each report) | | | $250.00 ($225.00, $312.50) | | | $250.00 ($225.00, $312.50) |
| Staff quality checking reports | 0.25 (0.13, 0.5) | $57.50[a] | $14.38 ($7.48, $57.50) | 0.25 (0.167, 0.5) | $57.50[a] | $14.38 ($7.48, $57.50) |
| **Provision of reports to each clinic: Labour and travel costs** | | | | | | |
| Staff labour costs involved in visiting and providing reports to clinic | 3.5 (1, 7) | $62.50[d] | $218.75 ($62.50, $437.50) | 3.5 (1, 7) | $62.50[d] | $218.75 ($62.50, $437.50) |
| Staff travel time to visit clinic (labour) | 6 (3, 17) | $62.50[d] | $375.00 ($187.50, $1,062.50) | 2 (0.5, 4) | $62.50[d] | $125.00 ($31.25, $250.00) |
| Flights, vehicle/parking expenses, and accommodation expenses to visit clinic | | | $421.50 ($259.00, $502.00) | | | $50.00[e] |
| **Total and incremental costs** | | | | | | |

(*Continued*)

**Table 3.** (Continued)

| Variable | ACCEPt-able costs (rural clinics) | | | Estimated costs for metro clinics (sensitivity analysis) | | |
|---|---|---|---|---|---|---|
| | Average number of hours used per clinic per quarter (range) | Hourly cost (in AUD) | Average total cost (AUD) per clinic per quarter (range) (unless otherwise indicated) | Average number of hours used per clinic per quarter (range) | Hourly cost (in AUD) | Average total cost (AUD) per clinic per quarter (range) (unless otherwise indicated) |
| Total costs per clinic per quarter to provide audit and feedback | | | $1,279.63 | | | $658.13 |
| Total costs for 27 clinics per year to provide audit and feedback | | | $138,200 | | | $71,078 |
| Total reduction in costs for removal of audit and feedback | | | −$138,200 | | | −$71,078 |
| Total reduction in costs per clinic per year for removal of audit and feedback | | | −$5,118.52 | | | −$2,632.52 |
| Number of patients in the target population in the 27 clinics where audit and feedback activities were removed[b] | | | 23,510 | | | 23,510[f] |
| Average saving per patient per year in the target population through removal of audit and feedback activities | | | −$5.88 | | | −$3.02[f] |
| Incremental change in proportion of target patients tested through removal of audit and feedback activities (95% CI)[g] | | | −3.1% (−6.2%, −0.1%) | | | −3.1%[f] (−6.2%, −0.1%) |
| Incremental cost of audit and feedback activities per additional patient per year in the target population tested (range) | | | $189.64 ($94.82, $5,117.49) | | | $97.42[f] ($48.71, $2,628.37) |

[a]Hourly rate is based on the hourly salary (AU$46) of a junior academic researcher plus 25% on-costs.

[b]See Table 2 for data.

[c]See Table 2 for results. The *p*-value for the incremental change in proportion tested was 0.1852, so incremental cost not calculated.

[d]Hourly rate is based on the hourly salary (AU$50) of a postdoctoral academic researcher plus 25% on-costs.

[e]Visiting metropolitan clinics would incur vehicle/parking costs of $50 per trip.

[f]Applying data from ACCEPt-able clinics to a metropolitan setting.

[g]See Table 2 for results. The *p*-value for the incremental change in the proportion tested was 0.0270.

AUD, Australian dollars.

'calibration' of the intervention and its support may have contributed to declining testing rates across all groups [38]. In addition, our intervention targeted GPs, with negligible patient involvement, which is necessary for sustaining change over time [39]. Nonetheless, ACCEPt-able highlights the challenges of sustaining GP behaviour change; further research is needed on how to sustain such change.

Several studies have reported on the removal of incentives in primary care, but all present observational data only, with conflicting results. Two studies examined incentive removal from the UK Quality and Outcomes Framework [5,14]. Similar to our findings, Kontopantelis et al. found that incentive removal had minimal effect on activities related to treatment and monitoring (e.g., cholesterol) [14]. In contrast, Minchin et al. found immediate reductions following incentive removal [5]. However, reductions were greatest where the GP was required to record advice provided to the patient (e.g., contraception advice) and smaller for activities related to measurement (e.g., cholesterol) [5,14]. Similar findings were observed in another study of 35 Kaiser Permanente facilities in the US, where small decreases in screening for diabetic retinopathy and cervical screening were observed when incentives were removed [12]. A cluster RCT of an intervention that included incentives to reduce high-risk prescribing in 34

primary care clinics in Scotland [13] found no change in high-risk prescribing during a 4-year observational post-intervention study when incentives were removed.

We are unaware of any RCT evidence about the impact of removing audit/feedback on provider activity. Observational data collected at the end of RCTs of audit/feedback interventions show similar results. An RCT of an intervention that included an educational session and audit/feedback found a 50% reduction in inappropriate antibiotic prescribing in 18 community-based paediatric clinics in the US, but once the intervention was terminated at trial end, there was an immediate increase in inappropriate prescribing, which returned to pre-trial levels within 18 months [40]. Similar findings were reported at the conclusion of another US trial of audit/feedback to reduce inappropriate prescribing [41].

Our trial has several limitations. First, our sample size assumed an absolute reduction in testing of 5% when incentives and/or audit/feedback were removed and no change where they were retained. We did not anticipate a decrease in all groups. However, the factorial design and smaller ICC than estimated (0.015 versus 0.02) maximised our statistical power. Second, when designing the trial, we assumed no interaction between removal of incentives and removal of audit/feedback and were not powered to detect an interaction. However, our post hoc analysis of each intervention group separately showed similar results to our primary analysis, confirming the factorial analysis findings. Third, 4 clinics did not provide testing data and were excluded from the analysis after randomisation. However, their removal had little impact on the distribution of minimisation and socio-economic variables across the intervention groups, and these variables were adjusted for in our analysis, minimising any bias (S3 Table). Fourth ACCEPt-able was undertaken in rural areas, so the results might not be generalisable to urban areas. However, our analysis accounted for cluster-level socio-economic factors, which had little impact on results. Fifth, we assessed the impact of the intervention on chlamydia testing in year 2 compared with baseline, and not all clinics remained in the trial until the end of year 2. However, it was reassuring that the average duration of the intervention period was similar between groups. Sixth, we evaluated the impact of the removal of incentives and audit/feedback on chlamydia testing, so our results may not be generalisable to other preventive health activities in general practice. Finally, this trial was set in Australia, where general practice is mainly renumerated on a fee-for-service basis; our results may be less transferrable to settings where incentives represent a larger proportion of income.

## Conclusions

In this cluster RCT, we found that the financial incentives offered had little impact on chlamydia testing in Australian general practice. The total amount of financial incentive payments received per year in our trial was consistent with other incentive payments GPs received at the same time in Australia. It is possible that the removal of financial incentives might have a greater impact if incentive payments made up a greater proportion of GP income, such as in the UK. RCT evidence is needed to investigate this question. The removal of audit and feedback with a face-to-face visit resulted in a relative reduction in testing activity of 30% overall. A reduction of this size could have a considerable public health impact at the population level, with fewer chlamydia tests conducted and more infections going undetected. Our results suggest that, in Australia at least, audit and feedback is an important intervention for influencing GP behaviour for preventive health activities like chlamydia testing. The use of digital platforms that include automated reports and online communication could reduce the costs associated with audit and feedback. Our finding that chlamydia testing also decreased in clinics that retained incentives and audit and feedback highlights that simply retaining these

interventions over time is not enough; further studies should investigate how to sustain clinician behaviour change over time.

## Supporting information

**S1 CONSORT Checklist.**
(DOCX)

**S1 Fig. Annual chlamydia testing rates for ACCEPt and ACCEPt-able.**
(PDF)

**S1 Table. The primary outcome, chlamydia testing, by sex and age group: Factorial analysis.**
(DOCX)

**S2 Table. The primary outcome, chlamydia testing: Intervention group analysis.**
(DOCX)

**S3 Table. Distribution of minimisation and socio-economic status variables across clinics by intervention group.**
(DOCX)

## Acknowledgments

The authors would like to thank the ACCEPt-able research officers for their efforts in implementing and supporting clinics during the trial; participating clinics, GPs, and nurses; Associate Professor Douglas Boyle and the Health Informatics Unit, Department of General Practice, University of Melbourne, for ongoing technical support with regards to data extraction from medical records software; Professor Jane Tomnay from the Centre for Excellence in Rural Sexual Health, University of Melbourne, for ongoing guidance around conducting research in rural areas; and pathology providers for their contribution to data collection.

The views expressed are those of the authors alone and do not necessarily reflect those of the funding body.

## Author Contributions

**Conceptualization:** Jane S. Hocking, Meredith Temple-Smith, Matthew Law, Mieke van Driel, Christopher K. Fairley, Basil Donovan, Rebecca Guy, Nicola Low, John Kaldor, Jane Gunn.

**Data curation:** Jane S. Hocking, Sabine Braat.

**Formal analysis:** Sabine Braat, Matthew Law, Liliana Bulfone, Callum Jones.

**Funding acquisition:** Jane S. Hocking, Meredith Temple-Smith, Matthew Law, Liliana Bulfone, Mieke van Driel, Christopher K. Fairley, Basil Donovan, Rebecca Guy, Nicola Low, John Kaldor, Jane Gunn.

**Investigation:** Jane S. Hocking, Anna Wood.

**Methodology:** Jane S. Hocking, Sabine Braat, Matthew Law, Liliana Bulfone, Christopher K. Fairley, Basil Donovan, Jane Gunn.

**Project administration:** Anna Wood.

**Supervision:** Jane S. Hocking, Anna Wood, Meredith Temple-Smith.

**Writing – original draft:** Jane S. Hocking, Anna Wood, Sabine Braat.

**Writing – review & editing:** Jane S. Hocking, Anna Wood, Meredith Temple-Smith, Sabine Braat, Matthew Law, Liliana Bulfone, Callum Jones, Mieke van Driel, Christopher K. Fairley, Basil Donovan, Rebecca Guy, Nicola Low, John Kaldor, Jane Gunn.

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
