## [Editor Report · Decision Letter 0]

9 Jun 2021

Dear Dr Hocking, 

Thank you for submitting your manuscript entitled "The impact of removing financial incentives and/or audit and feedback on preventive care activities in general practice: A cluster randomised controlled trial (ACCEPt-able)" for consideration by PLOS Medicine.

Your manuscript has now been evaluated by the PLOS Medicine editorial staff and I am writing to let you know that we would like to send your submission out for external peer review.

Please re-submit your manuscript within two working days, i.e. by Jun 11 2021 11:59PM.

Kind regards,

Beryne Odeny

Associate Editor

PLOS Medicine

---

## [Decision Letter · Decision Letter 1]

11 Aug 2021

Dear Dr. Hocking,

Thank you very much for submitting your manuscript "The impact of removing financial incentives and/or audit and feedback on preventive care activities in general practice: A cluster randomised controlled trial (ACCEPt-able)" (PMEDICINE-D-21-02501R1) for consideration at PLOS Medicine. 

Your paper was discussed among the editors and sent to independent reviewers, including a statistical reviewer. The reviews are appended at the bottom of this email and any accompanying reviewer attachments can be seen via the link below:

[LINK]

In light of these reviews, we will not be able to accept the manuscript for publication in the journal in its current form, but we would like to invite you to submit a revised version that addresses the reviewers' and editors' comments fully. You will appreciate that we cannot make a decision about publication until we have seen the revised manuscript and your response, and we expect to seek re-review by one or more of the reviewers. 

We hope to receive your revised manuscript by Aug 31 2021 11:59PM. Please email us (plosmedicine@plos.org) if you have any questions or concerns.

Please let me know if you have any questions, and we look forward to receiving your revised manuscript. 

Sincerely,

Richard Turner PhD, for Beryne Odeny

Senior editor, PLOS Medicine

rturner@plos.org

Noting PLOS' data policy, https://journals.plos.org/plosmedicine/s/data-availability, please state whether the study's ethics approval would permit the study data to be shared with other researchers under conditions of confidentiality.

In the abstract and throughout the paper, please quote p values alongside 95% CI, where available.

Please add a new final sentence to the "Methods and findings" subsection of your abstract, which should begin "Study limitations include ..." or similar and should quote 2-3 of the study's main limitations.

Please relocate the Author Summary after the Abstract. 

Throughout the text, please remove spaces from within the square brackets for reference call-outs (e.g., "... on others [4,9,10].").

Please remove the information on competing interests and funding from the end of the main text. In the event of publication, this will appear in the article metadata, via entries in the submission form. 

Please abbreviate journal names consistently in your reference list.

Please add a completed checklist for the appropriate CONSORT extension as a supplementary document, labelled "S1_CONSORT_Checklist" or similar and referred to as such in the Methods section. 

In the checklist, please refer to individual items by section (e.g., "Methods") and paragraph number, not by line or page numbers as these generally change in the event of publication 

Comments from the reviewers:

*** Reviewer #1: 

Hocking and colleagues submitted an interesting and well-written manuscript on the impact of removing financial incentives and/or audit/feedback on chlamydia testing rates in Australian GP clinics. They conducted a cluster randomized controlled trial to address this question, which to date has only been addressed by a few observational studies. They find declining testing rates over time for all groups, but only removal of audit/feedback resulted in a reduction of these rates beyond the existing trends. The manuscript contributes meaningfully to filling an important knowledge gap in this field. The authors appear to have done a very good job in designing and executing their study, and in reporting their results in this manuscript. I have read the manuscript carefully, and only have a few minor comments, which are detailed below:

1. The authors mention in the discussion that incentives (and perhaps audit/feedback as well) may not have been critical in driving test uptake. If that is the case, why would you then expect a decrease in preventive activities if these were removed (see hypothesis in p. 4, l. 118). It would be helpful if this authors would motivate their hypothesis more based on what is known from previous work in Australia and elsewhere.

2. In the discussion the authors mention that the incentives comprised less than 10 percent of GP practices' funding. It would be helpful if this would be mentioned earlier, e.g. in the section 'study design and participants'. In addition, what is the maximum incentive size relative to GP's maximum income (or GP practices revenues)? That would tell the reader something about the actual magnitude of the incentive.

3. A general point is that the authors could be a bit more consistent in the use of the terms 'GP' and 'Clinic', e.g. on page 5. It is not always clear if the authors talk about individual GPs or about the clinics that they work in. 

4. p. 6, l 157: why the choice for the 67th percentile?

5. p. 6, l 170-171: I wonder why the authors used this as the denominator instead of all patients in this age range. Perhaps this has something to do with the fact that GPs in Australia are not paid on a capitation basis (?) and only on a FFS basis, which would imply that they only 'observe' their patients when they present themselves with a health issue?

6. p. 10, l 283: given that the focus was on Chlamydia screening rates, I don't think the authors can say something about 'preventive activities' in general. I think this should be changed in the title of the manuscript as well.

7. p. 10, l 305-307: I found this part a bit difficult to follow.

*** Reviewer #2: 

This is a well designed and conducted 2X2 factorial cluster-RCT on the impact of removing financial incentives and/or audit and feedback on preventive care activities in general practice. The study design, outcomes, sample size, randomisation, trial registration, protocol, statistical methods and analyses, and presentation and interpretation of the results are mostly adequate. Especially, testing the assumption that there was no interaction effect between the two interventions was well done as the factorial design is only valid if there is no interaction between the interventions. However, there are still a few major issues needing attention.

1) In the statistical analysis section on page 7, it says "our analysis was a modified intention-to-treat as clinics that were unable to provide outcome data at trial end were excluded from the primary analysis". However, it's either ITT or not ITT, the wording 'modified ITT' is vague and widely criticised so should be avoided. This is essentially a complete case analysis as 4 clusters/clinics were dropped and excluded in the primary analysis.

2) Analyses. As 4 clusters/clinics were dropped and excluded in the analyses, the randomisation were broken and interupted at both cluster and patients levels, therefore the primary analyses only adjusted for two minimisation factors are not sufficient and inadequate as clearly we can see the imbalance at the cluster level in socioeconomic status. Instead, fully-adjusted analyses should be used for primary and all analyses throughout the paper to avoid potential bias due to the loss of 4 clusters.

3) Missing data. Normally, a complete case analysis will go alongside with a sensitivity analysis with missing data imputation. However, the missing data issue was not dealt with at all in this study. While, it may not be feasible in this study for missing data imputation as 4 clinics were dropped, but it would be very useful to compare the characteristics of dropped clusters with that of remaining clusters in the same arm to see whether it's missing at random or not at random so that make sure we are able to address the potential bias and impact of the missing data on the trial results.

*** Reviewer #3: 

Summary

The original ACCEPt Trial was of a multifaceted, clinic-based intervention using computerized reminders, an education package, financial incentives, and feedback on testing rates. The ACCEPt trial resulted in a significant increase in Chlamydia testing rates of eligible patients between control and intervention practices (13% versus 20%), but there was no difference in the primary outcome, the prevalence of chlamydia among patients aged 16-29 who attended the clinics "at the end of the intervention period" (2.5 to 4 years later).

In the current manuscript, "The impact of removing financial incentives and/or audit and feedback on preventive care activities in general practice: a cluster randomised controlled trial (ACCEPt-able)," Hocking and colleagues report on an RCT of the removal of either the financial incentive, audit and feedback, both, or neither on the rates of Chlamydia testing in primary care practices that were part of the original ACCEPt intervention group. Of 59 randomized practices, 55 contributed data.

General Comments

This appears to be a revision. I was not one of the initial reviewers.

This is a potentially interesting analysis about a randomized deimplementation of financial incentives and audit and feedback following their successful implementation (in regards to increasing testing rates). 

I have several major concerns with the manuscript and analysis.

First, the analytic method, the authors' own definition of clinical significance, and the marginal nature of the results call the conclusions into question. 

The protocol said that "analyses will be adjusted for the chlamydia testing rate at each general practice immediately prior to commencing ACCEPt-able" and "account for cluster…GP and patient variability." In the actual analysis, the investigators adjusted for clinic clustering, annual chlamydia testing rates, and the number of 16 to 29-year-olds attending the clinic each year. A more fully-adjusted model included patient sex, age-group, and socio-economic status of the clinic. Only the less-adjusted model examining the impact of the removal of audit and feedback was of marginal statistical significance, and even here, the odds ratio - which the authors identify as the planned primary analysis (page 7, line 202) - includes 1.0. Further adjustment yielded a non-significant result. 

In the Methods, the investigators cited a 5% absolute reduction as a "clinically relevant result." (I have actually used this exact clinically significant difference in some of our own analyses of quality or health services research.) The absolute reduction was only a 3.1% absolute decline. 

In all, this does not seem strong enough to hang the conclusion that "financial incentives don't work, but audit and feedback does so primary care practices should invest in audit and feedback." (See "Third" below regarding overgeneralizing the result.)

Second, crucial details of GPs workflow around chlamydia testing are not described that might help the reader understand the lack of effectiveness of either the ACCEPt intervention or the persistence of financial incentives or audit and feedback. The reader is left to wonder what it is about ordering Chalmydia testing that is so difficult that an intensive, multifaceted intervention only led to a 7% increase in testing rates and those rates reverted almost back to their pre-ACCEPt level. For behavioral interventions, details of workflow and interventions are extremely important (Fox et al. BMJ. 2020;370:m3256), perhaps especially when they are not successful. It is overly simplistic to say "financial incentives and audit and feedback don't work." 

Regarding chlamydia testing, how, by whom, and when was it done? Regarding the financial incentives, in what context were they delivered (i.e., was it an unrecognizably small part of some larger, quarterly payment or was it identifiable as a "chlamydia testing incentive"). Regarding the feedback, all we know is that it was "given to GPs during a [quarterly] visit with a research officer," but how was it delivered? Was it provided with a descriptive norm or injunctive norm? Lack of these details makes it hard for readers to learn anything from the ACCEPt and ACCEPt-able experience.

I am also curious about the details of how financial incentives and audit and feedback were deimplemented. Practices were invited and had to consent to participate in this deimplementation RCT. As such, presumably they were told they were going to have a prior intervention removed ("A research officer informed clinics of their allocation"), but how was this done? Thus, the invitation, consent, and enrollment process must have included the implicit message that "Chlamydia testing is less important than it was before" or "we are not going to be monitoring Chlamydia testing as closely." Just as the Hawthorne effect in an intervention trial is often the most impactful part of the interventions, as part of deimplementation, these implicit messages could have had as big an effect as the actual removal of the interventions. Indeed, most of the decrease in Chlamydia testing occurred between the Baseline and Year 1. Also, what happened to the computerized alerts? Did they persist?

With apologies for 1 paragraph of editorializing, given the findings of ACCEPt and ACCEPt-able, I would guess the clinic environment and the mechanism of ordering was reliant on busy GPs to remember to bring up, discuss, and order Chlamydia testing that was not integrated into regular work-flow. Financial incentives and audit and feedback have their place in nudging clinicians regarding complex decision-requiring behaviors. But for something as simple as Chlamydia testing, a much better solution is to use practice facilitation to systematize or routinize the activity and remove it from the need for the GP to remember and act. In my own health system, we achieve rates around 90% when we have our check-in or rooming staff systematically perform tasks and use standing orders like for influenza vaccination, fall screening, depression screening, and tobacco screening. 

Third, related to the marginal nature of the results and details around practice and interventions, it is an overgeneralization to say that these interventions did or did not affect "preventive care in general practice." Because the details of workflow are crucial, all that has been shown is that "the removal of financial incentives or audit and feedback did not affect Chlamydia testing in rural Australian general practices." While that might be overly narrow, the details around individual preventive services, like Chlamydia testing, cannot be generalized to ALL preventive services (e.g., counseling, cancer screening, immunizations, etc.).

Fourth, given that the interventions were delivered on a quarterly basis, it is unclear why the investigators chose to analyze their data on an annual basis and, in the analysis, only compare Year 2 to the baseline year. This effectively ignores all the data from Year 1 (when more practices were participating) and treats all of the data collected during Year 2 as equivalent (i.e., a visit in Month 13 is the same as a visit in Month 14). It also ignores the possibility that changes in Chlamydia screening among the groups could have had different trajectories and could have overweighted the contribution of clinics that participated in the trial longer (the analysis was adjusted for clustering by practice, but not by practice volume).

Specific Comments

Title: As noted above, including "preventive care activities" in the title regarding an intervention that had to do with Chlamydia screening is an overgeneralization.

Page 3, Line 66: Here, and throughout the manuscript, the investigators never state the number of GPs or the number of GPs clustered within practices. I see there were 305 GPs in the 63 clinics randomized to the interventions in the original ACCEPt Trial.

Page 3, Line 71: More a problem with presentation, here and elsewhere, it is confusing that the authors present the RCT as organized in 4 groups, but then only analyze the data in 2 groups. This can be improved by mentioning the analytic plan earlier on and, in Lines 77 through 84, introducing the actual comparison groups much earlier in the sentences. As written, one only discovers the group being discussed in the middle of the sentences, after some data about those groups has been presented.

Page 3, Abstract General Comment: The timeframe of the assessment of the primary outcome is not stated (in Year 2 after randomization). 

Page 5, Line 123: The authors say there were no changes to the trial methods, but later report a protocol deviation (modified intention-to-treat analysis). 

Page 5, Line 136: The authors mention, in addition to financial incentives and feedback, computer alerts, but what about the educational package? That was part of the initial multicomponent ACCEPt intervention. Was that only done at the beginning of ACCEPt and never repeated?

Page 5, Line 143: If a patient reported risk factors or genital symptoms and required further testing, how were repeat patients counted? How were they handled in the analysis?

Page 6, Line 168: The authors state that the primary outcome was the "annual chlamydia testing rate," but do not say when this was assessed. It is not until the reader gets to Table 2 does it become clear that the primary analysis is between Baseline and Year 2.

Page 8, Line 243: The authors need to define the interpretation and direction of the interaction effect. Was the 3.2% (NS) the marginal increase in screening in practices randomized to retain both of the interventions relative to practices retaining neither? How is this conceptually different from the analysis presented on page 9, line 260?

Page 9, Line 274: Given that all of the clinics participating in ACCEPt and ACCEPt-able were rural, it is not clear how information about metropolitan clinics is included.

Page 18, Table 1: The table should include information about GPs within practices (central tendency and variation). 

Page 20, Table 2: In the footnote and elsewhere, the authors say the denominator is "N=number of individuals attending the clinic," but I think it should be the number of 16-29-year-olds. 

Page 21, Table 3: The rows about "authorising payments" are unclear. What do these represent?

***

[LINK]

---

## [Decision Letter · Decision Letter 2]

26 Oct 2021

Dear Dr. Hocking,

Thank you very much for re-submitting your manuscript "The impact of removing financial incentives and/or audit and feedback on chlamydia testing in general practice: A cluster randomised controlled trial (ACCEPt-able)" (PMEDICINE-D-21-02501R2) for review by PLOS Medicine.

I have discussed the paper with my colleagues and the academic editor and it was also seen again by two reviewers. I am pleased to say that provided the remaining editorial and production issues are dealt with we are planning to accept the paper for publication in the journal.

[LINK]

We look forward to receiving the revised manuscript by Nov 02 2021 11:59PM.   

Sincerely,

Beryne Odeny, 

PLOS Medicine

plosmedicine.org

Requests from Editors:

1) The Data Availability Statement (DAS) requires revision. If part of the data is not freely available, please include an appropriate contact (web or email address) for inquiries (this cannot be a study author/ co-author).

2) Please place the Author Summary after the Abstract.

3) Abstract - In the last sentence of the Abstract Methods and Findings section, please describe the main limitation(s) of the study's methodology. 

4) The terms gender and sex are not interchangeable (as discussed in http://www.who.int/gender/whatisgender/en/ ); please use the appropriate term.

5) Please indicate in the figure captions the meaning of the bars and whiskers in the figures

6) Please remove the information on funding acquisition from the end of the main text. In the event of publication, this will appear in the article metadata, via entries in the submission form

7) References: 

a) Please ensure there is no space between in-text reference call outs. For example, “…community [2,8,9].”

b) Please ensure that journal name abbreviations consistently match those found in the National Center for Biotechnology Information (NCBI) databases. https://journals.plos.org/plosmedicine/s/submission-guidelines#loc-references. 

8) To help us extend the reach of your research, please provide any Twitter handle(s) that would be appropriate to tag, including your own, your coauthors’, your institution, funder, or lab.

Comments from Reviewers:

Reviewer #2: Many thanks authors for their great effort to improve the manuscript. I am mostly satisfied with the response and revision. However, one minor issue still remains. In response to my comments on missing data, the authors said "We have added in two additional supplementary tables...(Supplementary Tables 3A and 3B)". However, these two tables have neither been mentioned in the final clean version nor appeared anywhere in the supplementary information. Could authors please add and link these two supplementary tables in the submission? Also, it seems a previous version rather than a clean version was presented in the resubmission. Can authors make sure all the changes are included and appear in the final clean version?

Reviewer #3: This is a revised version. The "clean" version that was uploaded appears to be the same as the original (R1) revision. I am reviewing the response letter and the "marked" version. Not having the clean version has made this review more challenging. 

The authors have addressed most, but not all, of the prior critiques. In particular, the addition of details about the clinic environment and the nature of the intervention greatly improves the usefulness of the manuscript. It is good that the authors have made the language more specific about their intervention being limited to Chlamydia testing. 

The authors were non-responsive to my first comment about using multiple models and the lack of statistical significance of the results as shown in Table 2 (the OR for all treatment effects include 1.0).

With apologies, for my fourth point about the analysis only comparing year 2 in aggregate to the baseline year, included a typo. I meant to write that "a visit in Month 13 is the same as a visit in Month 24" (not "Month 14"). The authors were non-responsive on this point. At a minimum, they need to address this as a limitation.

[LINK]

---

## [Editor Report · Decision Letter 3]

2 Nov 2021

Dear Dr Hocking, 

On behalf of my colleagues and the Academic Editor, Dr. David Peiris, I am pleased to inform you that we have agreed to publish your manuscript "The impact of removing financial incentives and/or audit and feedback on chlamydia testing in general practice: A cluster randomised controlled trial (ACCEPt-able)" (PMEDICINE-D-21-02501R3) in PLOS Medicine.

PUBLICATION SCHEDULE

Given our busy publication schedule for the remainder of 2021, we are planning to publish your paper in early January 2022 (the exact date will be communicated to you once confirmed).

PRESS

Sincerely, 

Beryne Odeny 

PLOS Medicine